# Gender Differences in Dietary Patterns and Eating Behaviours in Individuals with Obesity

**DOI:** 10.3390/nu16234226

**Published:** 2024-12-06

**Authors:** Alessandra Feraco, Andrea Armani, Stefania Gorini, Elisabetta Camajani, Chiara Quattrini, Tiziana Filardi, Sercan Karav, Rocky Strollo, Massimiliano Caprio, Mauro Lombardo

**Affiliations:** 1Department of Human Sciences and Promotion of the Quality of Life, San Raffaele Open University, Via di Val Cannuta, 247, 00166 Rome, Italy; alessandra.feraco@uniroma5.it (A.F.); andrea.armani@uniroma5.it (A.A.); stefania.gorini@uniroma5.it (S.G.); elisabetta.camajani@uniroma5.it (E.C.); q.chiara96@gmail.com (C.Q.); tiziana.filardi@uniroma5.it (T.F.); rocky.strollo@uniroma5.it (R.S.); massimiliano.caprio@uniroma5.it (M.C.); 2Laboratory of Cardiovascular Endocrinology, San Raffaele Research Institute, IRCCS San Raffaele Roma, Via di Val Cannuta, 247, 00166 Rome, Italy; 3Department of Molecular Biology and Genetics, Çanakkale Onsekiz Mart University, Canakkale 17000, Türkiye; sercankarav@comu.edu.tr

**Keywords:** obesity, gender differences, dietary behaviours, eating frequency, hunger timing, body composition, principal component analysis, fat mass, structured eating

## Abstract

Background/Objectives: Obesity is a global health problem with significant chronic disease risks. This study examined gender differences in eating behaviour, body composition, eating frequency and time of hunger in an Italian cohort with obesity (BMI ≥ 30) to inform gender-specific management strategies. Methods: A retrospective analysis of 720 adults (51.5% female, mean age 44.4 ± 13.8 years) assessed body composition and eating behaviour using principal component analysis (PCA) to classify eating profiles (structured, irregular, social and disordered/impulsive eaters). Results: Males showed higher weight, abdominal circumference and fat mass, while females showed higher fat mass percentages (*p* < 0.001). Gender differences were observed in the frequency of meals (e.g., 54.7% of males and 64.7% of females consumed 4–5 meals per day, *p* = 0.0018) and the time of hunger (males: before dinner; females: morning hunger, *p* = 0.005). The PCA profiles revealed that the ‘structured eaters’ had a healthier body composition, whereas the ‘disordered/impulsive eaters’ had a higher fat mass. Irregular eaters were predominantly male (41.0%), while disordered eaters were predominantly female (39.9%) (*p* = 0.0016). Conclusions: Gender-specific eating patterns influence obesity outcomes. Structured eating was associated with healthier profiles, whereas impulsive or irregular patterns were related to higher fat mass. The retrospective design and non-validated questionnaire for dietary behaviour assessment limit generalisability, warranting further research for tailored interventions. Registration: ClinicalTrials.gov (NCT06654674).

## 1. Introduction

The global prevalence of obesity is steadily increasing and represents a critical public health challenge [1]. Gender differences in eating behaviour and metabolic outcomes underscore the need for tailored interventions to effectively address obesity. Recent systematic reviews and meta-analyses have demonstrated that hormonal and psychological factors mediate these gender-specific patterns [2,3].

A growing body of research suggests that eating behaviour is not uniform across populations and may differ according to gender, age and socioeconomic factors [4,5]. Gender, in particular, has been identified as a key determinant of eating behaviours with studies showing that men and women often have different eating patterns and preferences [6,7]. For example, women are more likely to report eating smaller, more frequent meals, while men tend to consume larger portions and engage in more irregular eating behaviours, such as skipping meals [8,9]. These differences may have important implications for interventions aimed at preventing or managing obesity, as they suggest that gender-specific strategies could improve the effectiveness of dietary recommendations [10].

Principal component analysis (PCA) is a widely used statistical method to identify patterns in multidimensional dietary data. In the context of dietary behaviour, PCA has been applied to examine how factors, such as food preferences, mealtimes and social eating habits, cluster to form distinct behavioural profiles [11,12]. These typologies can provide valuable insights into the diversity of eating behaviours within a population and help identify groups at high risk for obesity and other metabolic disorders. While previous reviews have examined gender differences in eating behaviour, few studies have employed PCA to investigate gender-specific behavioural profiles in obesity populations. This study applies PCA to a population with obesity, filling an important gap in the literature.

Despite growing interest in gender differences in eating behaviour, studies using multivariate approaches such as PCA remain scarce. This study aims to clarify how gender influences eating behaviour patterns in an Italian population with obesity. The results are intended to inform the development of more targeted and effective dietary interventions tailored to behavioural profiles.

## 2. Methods

The study is retrospective and involved a demographically heterogeneous cohort with participants from different regions of Italy. However, detailed information on the socioeconomic or regional diversity of the sample was not collected, which may limit the generalisability of the results. The sample size was determined on the basis of the number of eligible subjects within the obesity centre database. Out of a total of 2508 patients, 720 participants fulfilled the inclusion criteria (BMI ≥ 30, age 18–65 years, completion of an online questionnaire in Italian and written informed consent). This requirement may have introduced a selection bias, potentially excluding migrants, expatriates or individuals with lower digital literacy or limited knowledge of Italian. Only a few data points were missing, and these were handled using mean imputation. For the variables affected, the average of available values was used to replace the missing entries, while acknowledging that mean imputation may not fully eliminate bias in statistical analyses. The study protocol, including the consent form, received IRCCS San Raffaele ethics committee approval (registration number RP 23/13), adhering to the standards of the Declaration of Helsinki and its amendments. Registration on ClinicalTrials.gov (NCT06654674) was completed to ensure transparency and dissemination of the study results. Although not required for a retrospective design, this registration specifically concerns the study described and does not represent another clinical study.

### 2.1. Body Composition Analysis

Participants underwent a detailed medical evaluation, including dietary history, physical examination and body composition (BC) assessment. Measurements were taken after an overnight fast, with participants wearing minimal clothing. This process assessed their eating habits, physical health and body composition. For weight measurements, calibrated electronic scales with a capacity of up to 250 kg were used on a stable surface. The participants, barefoot and wearing light clothing, stood motionless on the scales while a clinical assistant recorded their weight to the nearest gram. Two measurements were taken for accuracy, with a third added if the first two differed by more than 100 g; the two closest values were used for analysis. Height was assessed with a stadiometer placed on a solid, flat surface. Participants stood barefoot with their heads aligned with the Frankfurt plane, their heels and backs gently touching the stadiometer, arms at their sides, legs straight and feet flat. Two height readings were obtained and a third if there was a deviation greater than 0.1 cm; the two closest measurements were used. Body composition, comprising fat mass (FM), fat-free mass (FFM) and total body water (TBW) in percent and in kilograms, was measured with the Tanita BC-420 MA bioimpedance meter (Tanita Corporation, Tokyo, Japan). Although validated against the BodPod (COSMED Srl, Albano Laziale, Italia) [13], the degree of agreement and potential limitations of the method are acknowledged and discussed in the limitations. This device, which allows measurements in a standing position without electrodes, provides an accuracy within 100 g. Prior to the evaluation, participants had to have been fasting for at least three hours, abstain from vigorous physical activity for 12 h and avoid excessive food, drink and alcohol for 12 h. Women were advised and monitored to avoid testing during their menstrual cycle to improve accuracy. However, no specific measures have been implemented to monitor compliance with this guideline, which could introduce variability into the measurements.

### 2.2. Survey

Prior to the initial visit to the medical centre, participants completed a detailed online questionnaire, accessible on any Internet-enabled device. The survey, lasting approximately 30 min, began with an option for participants to give or withhold consent before proceeding. The questionnaire, although not formally validated, was structured in four sections focusing on food preferences, meal frequency, hunger times and interest in sport. Its design was inspired by commonly used nutritional tests [14], with the aim of capturing culturally relevant and specific food behaviours and preferences of the Italian population affected by obesity. The rationale for the design of the questionnaire was to address specific eating behaviours related to obesity management, ensuring its relevance to the objectives of the study. The questions were formulated to be intuitive and simple, facilitating responses and maintaining consistency with the principles of dietary behaviour assessment. To preserve anonymity, all responses were recorded without personal identifiers. This questionnaire, structured in four sections, has been used in our previous studies [7,9], and this survey focused specifically on the sections on food preferences and interest in sport. Participants were asked to self-identify their gender through the question ‘What is your gender?’ with the response options ‘Male,’ ‘Female,’ and ‘Prefer not to say.’ In this study, all participants chose to identify as either male or female. This approach acknowledges the self-reported nature of gender identity while being inclusive of diverse gender identities. It is important to note that the questionnaire was neither formally validated nor adapted from previously validated instruments. This limitation is recognised as it may affect the reproducibility and reliability of the results.

Taste preferences were assessed through the question ‘Do you like the following foods?’ with three possible answers: ‘I like’, ‘I don’t like’ and ‘I don’t know’. Participants provided feedback on a range of foods, including cow’s milk, plant-based alternatives (e.g., soya milk), low-fat and low-sugar yoghurt, fresh cheese, various meats, processed meats (such as ham), fish, eggs, pulses, both cooked and raw vegetables, fruit, cereals (e.g., spelt, barley), whole-grain foods, nuts, tofu and dark chocolate with a cocoa content of more than 75%.

### 2.3. Principal Component Analysis (PCA)

Participants were categorised into four PCA groups—structured eaters, irregular eaters, social eaters and disordered/impulsive eaters—based on their answers to questions on eating behaviour. Principal component analysis was performed using varimax rotation to simplify the component structure and improve interpretability. The number of selected components was determined using Kaiser’s criterion, which selects components with eigenvalues greater than 1, ensuring that only those explaining more variance than a single variable are included. In addition, the scree plot was inspected to identify the ‘elbow point’, where the reduction in eigenvalues began to level off.

The structured eaters were characterised by regular meal patterns, early hunger during the day, no skipped meals and a tendency to eat with others while avoiding distraction or hurried eating. Irregular eaters showed disorganised eating habits, frequent skipping of meals and occasional episodes of uncontrolled eating (usually once a month), with hunger typically reported before dinner. Social eaters showed minimal meal skipping and distracted eating, with meals eaten mostly in social settings and hunger most frequently before dinner. Disordered/impulsive eaters showed irregular meal patterns, frequent episodes of uncontrolled eating (more than once a week), eating in a distracted or hurried manner, eating alone and hunger occurring predominantly before dinner.

These behavioural profiles show distinct patterns influenced by lifestyle and psychological factors. A detailed summary of the eating behaviours associated with each PCA group can be found in Appendix A.

### 2.4. Statistical Analysis

Descriptive statistics (mean ± SD) were calculated for continuous variables (e.g., age, weight, BMI and body composition metrics) and independent t-tests were used to assess gender differences. Categorical variables (e.g., smoking status, food preferences and PCA group distributions) were analysed with chi-square tests. PCA classified participants into four eating behaviour profiles. The PCA was conducted with varimax rotation to simplify the component structure and improve interpretability. Varimax rotation was chosen to achieve a simpler and more interpretable factor structure. Kaiser’s criterion (eigenvalues > 1) was used to determine the number of components to be retained and scree plots were examined to confirm the selected solution. Factor loadings greater than 0.4 were considered significant for the assignment of variables to components. One-way ANOVA was used to compare body composition metrics between the PCA groups, stratified by gender. Bonferroni correction was applied to control for multiple comparisons, ensuring robustness of statistical significance. To address potential confounding variables, exploratory analyses were conducted considering age and gender as covariates. However, the analysis did not include specific adjustments for medication use, comorbidities or physical activity levels, which is recognised as a limitation of the study. Analyses were conducted using SPSS v. 28 software (IBM Corp., Armonk, NY, USA). Statistical significance was set at *p* < 0.05 for all tests, with adjusted thresholds for multiple comparisons where relevant.

## 3. Results

In this analysis, several gender-based differences in demographic and anthropometric parameters were identified (Table 1). Age showed a slight but significant variation between males and females (*p* = 0.010), with females having a higher mean age. Weight, fat mass (higher in females), AC, FFM (higher in males), body water and BMR showed highly significant differences (all *p* < 0.001), indicating substantial gender variability in body composition. In contrast, no significant differences were observed in BMI (*p* = 0.615) or smoking status (*p* = 0.312).

The analysis showed a high prevalence of preference for products such as fresh cheese (94.0%), meat (95.2%) and fruit (89.5%) in the entire obese population (Table 2). Notable differences between males and females included the preference for cow’s milk and whole-grain foods, where females reported a higher percentage (79.7% versus 72.0%, *p* = 0.0314, and 94.6% vs. 89.6%, *p* = 0.0311). However, for most foods, including red meat, processed meat and fish, no statistically significant gender differences in taste preferences were found (*p* > 0.05).

The distribution of food frequency differed significantly between males and females (*p* = 0.0018). The most common food frequency category for both sexes was 4–5 times a day, for 54.7% of males and 64.7% of females. However, a significant higher percentage of males reported eating 1–3 times per day (39.3% vs. 27.5% of females), while females showed a slight increase in the 6+ times per day category compared to males (Figure 1).

Our data revealed significant gender differences in hunger time preferences at specific intervals of the day (Figure 2). For example, a higher proportion of males reported feeling hungry ‘before dinner’ (39.5%) than females (28.8%). In contrast, females reported feeling hungry ‘upon waking’ (5.7%) and ‘in the morning’ (26.4%) more than males, with 1.7% and 21.2%, respectively. Females exhibited a more evenly distributed hunger pattern, spanning from morning to before dinner. Overall, chi-square analysis confirmed that these differences were statistically significant (*p* = 0.0054).

The analysis revealed that the gender distribution between the PCA groups was statistically significant (*p* = 0.0016). Irregular eaters were the largest group, comprising 41.0% of males and 32.3% of females (Figure 3). In contrast, females predominated in the disordered/impulsive eaters group (39.9% vs. 28.1% for males). Social eaters and structured eaters are smaller groups overall, with a slightly higher proportion of females in social eaters and males in structured eaters (Figure 3).

The heatmap illustrates the percentage distribution of male and female participants within each PCA (principal component analysis) group. The *p*-value derived from the chi-square test of independence is 0.0016.

In the overall patient group, fat mass percentage differed significantly across PCA groups (*p* = 0.048), with disordered/impulsive eaters and irregular eaters exhibiting higher fat mass percentages compared to structured eaters. The observed effect size (η^2^) was 0.039, indicating a modest effect (Table 3). When analysing fat mass percentage by gender, these differences were not statistically significant in males (*p* = 0.192) or females (*p* = 0.448), suggesting that the effect may be more pronounced when not stratified by gender.

For fat-free mass (FFM), females displayed a significant difference across PCA groups (*p* = 0.041), with structured eaters showing the highest FFM values, suggesting that this group may have comparatively higher muscle mass than other groups (Table 3).

## 4. Discussion

Obesity, mainly caused by poor diet quality and energy imbalance, represents a critical public health challenge. Addressing gender-specific dietary behaviour is essential to improve adherence to nutritional recommendations and effectively manage obesity. This study highlights distinct gender-specific patterns in meal frequency and timing among individuals with obesity. The observed gender-specific eating patterns are in line with studies showing emotional and impulsive eating behaviour in obese populations and bariatric surgery candidates. For example, recent evidence emphasises the role of emotional regulation in shaping these behaviours, with women showing a greater tendency to eat emotionally and impulsively than men, as observed in studies of bariatric surgery candidates [15,16]. Men showed a greater tendency to eat fewer daily meals than women, a behaviour that may reflect differences in appetite control and gender-related food preferences [17]. These findings are consistent with previous studies indicating that males tend to prefer larger and less frequent meals, whereas females show a propensity for smaller and more frequent meals, potentially to better manage hunger throughout the day. Interestingly, a different correlation between meal frequency and micro- and macronutrient intake was observed between the two sexes, suggesting the difference in food preferences between males and females may influence eating patterns [18]. Our findings corroborate prior studies on gender-specific eating behaviours while identifying unique patterns, such as the higher prevalence of structured eating profiles among women. This distinction may reflect both physiological differences, such as hormonal influences on appetite, and sociocultural factors that encourage regular dieting in women [9]. Furthermore, the distribution of hunger times showed significant differences between genders, with males reporting hunger more frequently before dinner and females more often upon waking or in the morning. This pattern could result from physiological and hormonal influences that regulate appetite according to gender, as suggested by studies on the impact of circadian rhythms and hormonal fluctuations on appetite. Notably, the link between short sleep duration and obesity is more frequently reported in women, potentially due to hormonal and circadian influences [19,20]. These physiological differences, combined with sociocultural factors, highlight the complexity of gendered eating behaviours and their potential role in the development of obesity [20]. However, our study does not directly measure emotional regulation or psychological factors, which limits the possibility of drawing definitive conclusions on their role in the observed behavioural profiles. Future research incorporating psychological assessments is needed to better contextualise these findings. Our findings may form the basis for dietary interventions that take these gender-specific differences into account, with the aim of improving the adherence and effectiveness of nutritional recommendations for the management of obesity. Furthermore, addressing emotional regulation in women, as well as impulsive and irregular eating in men, may provide more targeted strategies to mitigate obesity-related risks. These findings underline the crucial need for personalised, gender-specific dietary interventions that address emotional regulation and impulsive eating behaviour, with the potential to significantly increase adherence and improve long-term outcomes of obesity management. Addressing emotional dysregulation and impulsive eating through tailored interventions could enhance adherence to dietary recommendations and improve long-term obesity outcomes.

The significant gender differences observed in hunger timing preferences suggest that males and females may experience or perceive hunger cues at different times, potentially reflecting variations in lifestyle, metabolic rhythms or hormonal influences. These findings are in line with previous research [21,22] indicating gender-specific patterns in food intake and hunger perception, potentially shaped by factors such as meal timing, physical activity and circadian alignment.

Compared to our previous studies [7,9], which included both obese and non-obese subjects, the exclusively obese sample of this study shows some differences in dietary behaviour and preferences. While finding similar gender trends, such as a greater male propensity to consume foods of animal origin and a female preference for cow’s milk, obesity seems to amplify or attenuate some of these inclinations. For example, whereas in previous studies gender differences in meat, fish and dairy consumption were apparent in non-obese subjects, in this obese sample, these differences were less pronounced for some specific foods, suggesting that obesity may influence food preferences regardless of gender [23,24]. Furthermore, the present obese sample shows a high and uniform preference for energy-dense foods, such as meat and cheese, with less variability between genders than the mixed samples. This reduced variability in food preferences may result from metabolic adaptations associated with obesity, which alter appetite regulation and drive a greater preference for energy-dense foods, potentially overriding individual or gender-specific differences observed in non-obese populations. Notably, comparison between obese women and men reveals a different neural activation in brain regions involved in cognition and emotions potentially driving gender-specific differences in eating behaviour [23,24]. Neuroimaging studies showing heightened reward and emotion regulation activity in women underscore the importance of addressing emotional dysregulation to mitigate impulsive eating and its associated metabolic risks [25].

The analysis of the PCA groups revealed significant gender differences in eating behaviour profiles. We recognise that the observed differences in fat mass percentages between the PCA groups (*p* = 0.048) correspond to a small effect size (η^2^ = 0.039). Although statistically significant, the small effect size observed for differences in fat mass percentages between PCA groups (η^2^ = 0.039) suggests limited clinical relevance. Future research should explore whether these differences translate into significant outcomes for obesity-related health risks. Women were more represented among the disordered/impulsive eaters, whereas men tended to be more frequently among the irregular eaters. These findings emphasise that gender differences in eating behaviour are also evident in PCA profiles, suggesting that psychological factors, such as emotional regulation, and social influences, including cultural norms related to eating, distinctly shape eating behaviour in men and women with obesity [26]. In the group of disordered/impulsive eaters, we observed that a higher percentage of women tend to fit this profile. These individuals present an eating pattern characterised by irregular meals, often associated with episodes of uncontrolled and distracted eating, which is reflected in higher fat mass percentages. Previous studies confirm that disordered eating behaviour is more frequent among women and is often related to greater impulsivity and emotional eating [27]. This type of behaviour is associated with increased fat mass and increased metabolic risk, confirming that eating impulsivity may exacerbate the risk of obesity and its complications [28]. Impulsivity, in particular, has been shown to predict greater fat mass and frequent episodes of emotional eating. Women, compared to men, often show stronger associations between impulsivity and maladaptive eating behaviour, underlining the need for personalised strategies to address this risk factor [29].

The group of irregular eaters, predominantly men in our study, is characterised by skipped meals and disorganised eating. The literature shows that the habit of skipping meals is often more prevalent among men, which may be related to intense work schedules or less attention to regularity of meals [30]. This behaviour, however, has been associated with increased visceral adiposity and an increased risk of metabolic syndrome, as irregular meals may negatively influence glycaemic control and insulin response [31]. The social eater group, in which we observed a more balanced gender distribution, tends to eat meals in company and rarely skips meals. Although less represented in terms of percentage than other groups, social eating behaviour has been associated with better weight control and healthier food choices, probably due to the positive influence of social contexts [32]. Previous studies indicate that eating meals in company may reduce the risk of impulsive eating and improve diet quality [33]. Finally, the group of structured eaters shows a prevalence of organised eating behaviour, with no skipped meals and a focus on regularity. In our sample, women in this group have a higher lean mass, suggesting an association between structured eating and better body composition. The literature supports the idea that a regular, planned diet contributes to better weight control and a reduction in the risk of obesity, especially among women, as it facilitates a more balanced distribution of energy throughout the day and promotes metabolic regulation [34]. This comparison highlights how different PCA profiles may influence body composition differentially, with disordered/impulsive and irregular eaters showing a greater tendency to accumulate fat, while structured eaters exhibit a healthier body profile. These observations suggest the importance of targeted interventions to promote structured eating habits, tailoring strategies according to the behavioural inclinations of each PCA group to improve obesity management and metabolic health [35].

This study has several limitations that must be acknowledged. Firstly, the lack of detailed data on the socioeconomic and regional diversity of the cohort limits the generalisability of the results to a wider population, despite participants coming from different Italian regions. The use of an online survey in Italian may have introduced a selection bias by underrepresenting migrants, expatriates and individuals with lower digital literacy or limited knowledge of Italian, potentially influencing the observed patterns of eating behaviour and their association with gender. The study relied on bioimpedance analysis (BIA) to measure body composition, a method that, although validated with the BodPod, lacks the accuracy of standard techniques such as dual-energy X-ray absorptiometry (DXA). Furthermore, no procedures were implemented to monitor compliance with menstrual cycle guidelines, introducing variability in measurement accuracy among female participants. Another major limitation is the use of an unvalidated online questionnaire to assess dietary behaviour and food preferences. Although adapted to the study population and aligned with established nutritional tests, the lack of formal validation and a standardised scoring system compromises the reliability and generalisability of the data. Future research should incorporate validated instruments adapted to the cultural and linguistic context of the population to improve the robustness and reproducibility of the results. The statistical analysis also had limitations, including the lack of adjustment for covariates such as medication use, comorbidities and physical activity, which are known to influence body composition and eating behaviour. Similarly, the PCA results, while robust, were not adjusted for additional covariates and future studies should consider alternative rotational methods and sensitivity analyses to validate the identified components. Furthermore, the cross-sectional design prevents causal inference between PCA-derived behavioural profiles and body composition metrics. Finally, some *p*-values (e.g., *p* = 0.048) reflect small effect sizes (η^2^ = 0.039). While our findings showed significant relative differences across groups, it is important to acknowledge that some of the absolute differences were relatively small. For example, the difference in the proportion of irregular eaters between males and females was statistically significant (*p* = 0.0016) but represented an absolute difference of only 8.7 percentage points. These modest differences, although statistically significant, may have limited clinical relevance and should be interpreted with caution.

## 5. Conclusions

This study revealed significant gender differences in the eating behaviour of individuals with obesity, as summarised in the Table 4. Men eat fewer meals, with hunger mainly before dinner, while women prefer frequent meals, with hunger more often in the morning. Disordered/impulsive eaters, mainly women, and irregular eaters, mainly men, show higher percentages of fat mass, while structured eaters, more commonly women, show higher lean mass, reflecting healthier body profiles. These findings emphasise the importance of personalised, gender-sensitive dietary interventions aimed at promoting structured eating habits. Such personalised strategies have the potential to increase adherence, address key metabolic risks and ultimately improve obesity management and related health outcomes.

## Figures and Tables

**Figure 1 nutrients-16-04226-f001:**
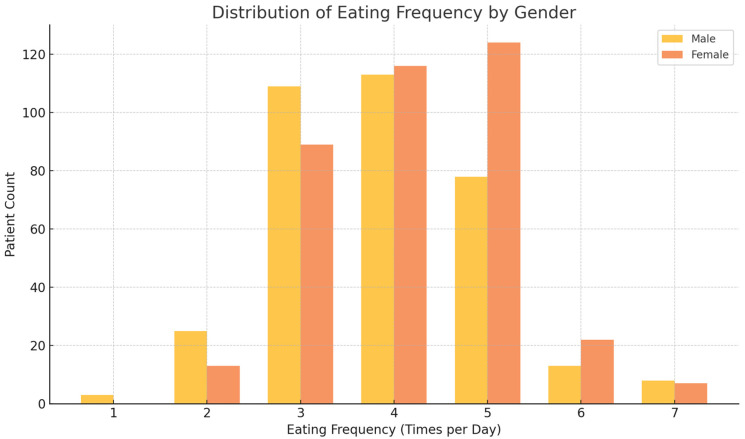
Distribution of eating frequency by gender in an obese population (BMI ≥ 30). Bar graph depicting the distribution of eating frequency (1–7 times a day) between males and females in an obese population. The chi-square test was used to assess the differences between the genders, with a significant value of 0.0018.

**Figure 2 nutrients-16-04226-f002:**
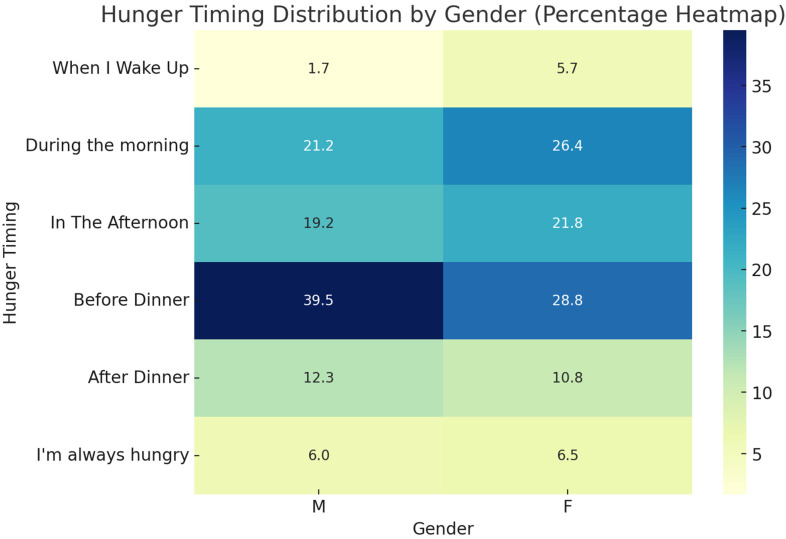
Hunger timing distribution by gender. This heatmap illustrates the percentage distribution of self-reported hunger times among male and female participants at different times of day. The categories include ‘When I wake up’, ‘During the morning’, ‘In the afternoon’, ‘Before dinner’, ‘After dinner’ and ‘I am always hungry’. Statistical differences between genders were assessed using the chi-square test, yielding significant results (χ^2^ = 16.58, *p* = 0.0054).

**Figure 3 nutrients-16-04226-f003:**
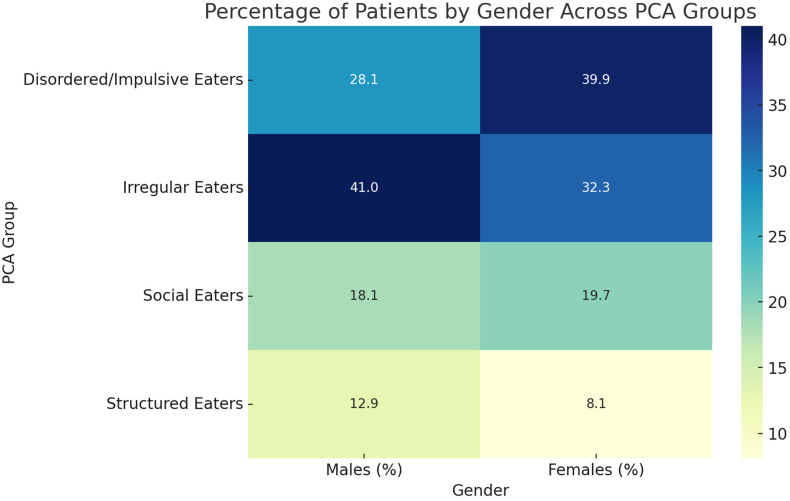
Percentage of males and females in each PCA group.

**Table 1 nutrients-16-04226-t001:** Descriptive statistics of study population by gender.

	Total	Total 95% CI:	M	M 95% CI	F	M 95% CI	*p*
n	720		349 (48.5%)		371 (51.5%)		
Age	44.4 ± 13.8	43.9–45.4	43.0 ± 13.6	41.6–44.5	45.7 ± 13.9	44.25–47.08	0.01
Smokers	21.8%		20.1%		23.5%		0.312
Weight	98.0 ± 16.4	96.81–99.20	105.7 ± 16.4	103.98–107.41	90.8 ± 12.6	89.49–92.06	<0.001
BMI	34.5 ± 4.2	34.17–34.78	34.4 ± 4.3	33.94–34.84	34.6 ± 4.1	34.13–34.97	0.0615
Fat Mass	37.0 ± 10.2	36.25–37.72	34.6 ± 10.5	33.53–35.72	39.2 ± 9.3	38.27–4 0.15	<0.001
Fat Mass (%)	37.4 ± 10.1	36.66–38.14	31.9 ± 10.3	30.83–32.99	42.6 ± 6.7	41.89–43.26	<0.001
AC	111.8 ± 10.4	111.08–112.58	115.1 ± 10.4	113.96–116.12	108.7 ± 9.4	107.87–109.76	<0.001
FFM	57.9 ± 11.9	57.02–58.75	67.6 ± 8.7	66.45–68.29	48.9 ± 5.9	48.35–49.57	<0.001
Body Water	43.9 ± 9.0	43.23–44.52	51.3 ± 6.2	50.37–51.68	37.1 ± 4.6	36.68–37.63	<0.001
BMR	1868.5 ± 368.8	1841.88–1895.09	2152.1 ± 290.9	2112.63–2173.60	1605.2 ± 199.3	1589.73–1630.56	<0.001

The table shows mean ± standard deviation (SD) of demographic and anthropometric variables across gender groups (M = males, F = females), with 95% confidence intervals (CIs) provided where applicable. Independent *t*-tests were applied for continuous variables (age in years, weight in kg, body mass index [BMI] in kg/m^2^, fat mass in kg, fat mass [%], abdominal circumference [AC] in cm, fat-free mass [FFM] in kg, total body water [TBW] in %, basal metabolic rate [BMR] in kcal/day), while the chi-square test was used for categorical data (Smokers [%]). Significant gender differences were observed in multiple parameters (*p* < 0.05), as indicated.

**Table 2 nutrients-16-04226-t002:** Dietary habit prevalence by gender in an obese population (BMI ≥ 30).

Food Item	% M	% F	% Total	*p*-Value M vs. F
Cow’s milk	72.0	79.7	76.0	0.0314
Vegetable drinks (e.g., soy milk)	35.3	37.1	36.3	0.7342
Low-fat white yoghurt	74.4	68.0	71.1	0.1345
Fresh cheeses	94.6	93.4	94.0	0.6108
Meat	95.5	95.0	95.2	0.9042
Red meat	94.1	89.1	91.8	0.1292
Processed meat (e.g., ham)	96.7	94.5	95.6	0.2236
Fish	94.1	93.8	93.9	0.9671
Eggs	96.5	93.0	94.7	0.0735
Legumes	97.2	94.8	95.9	0.1751
Cooked vegetables	96.8	97.5	97.1	0.7438
Raw vegetables	80.2	81.8	81.1	0.7004
Fruits	96.3	97.7	97.0	0.4213
Cereals (e.g., spelt, barley)	87.7	88.5	88.1	0.9116
Whole-grain foods	89.6	94.6	92.3	0.0311
Nuts	91.4	93.3	92.4	0.4489
Tofu	26.9	19.8	23.2	0.0885
Dark chocolate (at least 70%)	81.6	84.7	83.1	0.5116

Percentage of affirmative responses for certain eating habits among obese males and females (M = males, F = females). Chi-square tests were used to assess gender differences, with *p*-values for each comparison.

**Table 3 nutrients-16-04226-t003:** Body composition metrics by gender and PCA group.

Gender	Total	M	F	Total	M	F	Total	M	F	Total	M	F
PCA Group	Disordered/Impulsive Eaters	Disordered/Impulsive Eaters	Disordered/Impulsive Eaters	Irregular Eaters	Irregular Eaters	Irregular Eaters	Social Eaters	Social Eaters	Social Eaters	Structured Eaters	Structured Eaters	Structured Eaters
BMI	34.6 ± 4.3	34.5 ± 3.7	34.7 ± 4.6	34.1 ± 3.6	34.3 ± 3.9	33.9 ± 3.1	34.2 ± 3.3	33.7 ± 3.1	34.6 ± 3.5	33.6 ± 3.5	32.8 ± 3.0	34.6 ± 3.9
Fat Mass (%)	38.8 ± 8.4	31.5 ± 5.1	43.4 ± 6.8	37.6 ± 10.8	33.3 ± 12.6	42.6 ± 4.7	37.7 ± 6.8	32.7 ± 5.1	42.2 ± 4.6	35.5 ± 7.7	30.3 ± 4.7	43.2 ± 3.6
AC	111.5 ± 10.7	115.7 ± 10.4	108.9 ± 10.1	112.1 ± 10.6	115.3 ± 10.5	108.4 ± 9.6	111.4 ± 9.3	114.2 ± 9.2	108.8 ± 8.8	111.3 ± 8.2	112.9 ± 8.1	109.0 ± 7.8
FFM	56.5 ± 12.0	69.1 ± 7.8	48.7 ± 5.8	59.0 ± 12.6	67.8 ± 9.6	48.6 ± 6.1	57.5 ± 10.3	65.3 ± 8.2	50.5 ± 6.1	59.4 ± 12.0	67.7 ± 7.7	47.2 ± 3.9
Water	42.8 ± 9.1	52.3 ± 6.0	36.9 ± 4.6	44.8 ± 9.4	51.7 ± 6.7	36.9 ± 4.7	43.7 ± 8.0	49.9 ± 6.0	38.2 ± 4.9	44.5 ± 8.7	50.6 ± 5.1	35.6 ± 3.3
BMR	1819.9 ± 366.2	2185.4 ± 276.3	1591.7 ± 185.9	1906.0 ± 392.2	2168.5 ± 313.1	1600.1 ± 211.9	1848.1 ± 323.7	2072.3 ± 279.3	1646.9 ± 208.6	1889.6 ± 363.5	2128.7 ± 259.1	1538.8 ± 138.0

Body composition metrics across principal component analysis (PCA) groups, reported as mean ± standard deviation (SD). *p*-values represent the statistical significance of differences between PCA groups for each metric, analysed for total, male (M) and female (F) groups (calculated by analysis of variance [ANOVA]): body mass index [BMI] in kg/m^2^ (total: *p* = 0.173, M: *p* = 0.067, F: *p* = 0.396); fat mass [%] (total: *p* = 0.048, M: *p* = 0.192, F: *p* = 0.448); abdominal circumference [AC] in cm (total: *p* = 0.886, M: *p* = 0.421, F: *p* = 0.966); fat-free mass [FFM] in kg (total: *p* = 0.102, M: *p* = 0.072, F: *p* = 0.041); total body water [TBW] in % (total: *p* = 0.102, M: *p* = 0.091, F: *p* = 0.053).). Basal Metabolic Rate [BMR]. Detailed 95% confidence intervals (CIs) are provided in Appendix A.

**Table 4 nutrients-16-04226-t004:** Take-home messages.

Key Point	Main Message
Gender Differences in Meals	Obese men tend to eat fewer meals per day, while women prefer more frequent meals.
Timing of Hunger	Men report hunger primarily before dinner, while women experience hunger more in the morning.
PCA Behavioural Profiles	Disordered/impulsive eaters (more women) and irregular eaters (more men) show higher fat mass percentages.
Benefits of Structured Behaviour	Structured eaters, especially women, have higher lean mass, indicating an association with healthier profiles.
Energy Food Preferences	Both genders show a strong preference for energy-dense foods, such as cheese.
Importance of Customised Interventions	Dietary interventions should be tailored to gender and behavioural profiles to address metabolic risks and improve obesity management outcomes.

## Data Availability

The datasets produced and analysed during the present study are obtainable from the corresponding author upon reasonable request.

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
