# Peer review of "Gender Differences in Dietary Patterns and Eating Behaviours in Individuals with Obesity"

_nutrients, 2024, doi:10.3390/nu16234226_

Round 1
Reviewer 1 Report
Comments and Suggestions for Authors
This original article aims to characterize gender differences in eating behaviors among individuals with obesity. While the topic is clinically relevant, it is not particularly novel. Moreover, several significant methodological and formal flaws, along with a biased and incomplete discussion of the existing literature, substantially diminish the article's value, as outlined below.
Additionally, some sentences are void of meaning and substantial portions of the article appear to be composed using Chat-GPT or an equivalent software, leading to instances of vacuous, imprecise, or incorrect phrasing. It is paramount to:
1) openly acknowledge the utilization of Chat-GPT or an equivalent software;
2) meticulously review the entire manuscript with attention to wording, English language usage, and the intended meaning of the content. While AI tools may be used to enhance the manuscript's readability, the resultant content must be thoughtfully reviewed and aligned cohesively with the article's objectives and message.
The whole article needs to be revised to improve specificity and clarity of the terms.
All the points must therefore be addressed in a tracked-changed manuscript, before the article can be considered suitable for publication.
TITLE
The article is appropriate thank you.
ABSTRACT
There are two abstracts. I reviewed the second one, which is better, although with major flaws. The first one should be deleted and forgotten.
The abstract is poorly structured and more concise reporting of results and greater focus on the implications of findings would greatly enhance its readability and appeal to a broad audience.
Please include limitations in the abstract.
Sentences are lengthy and sometimes convoluted, which may hinder understanding.
The differences in eating frequency and timing of hunger are mentioned, but their implications are not fully contextualized. For example, how these behaviors influence obesity outcomes should be briefly discussed.
PCA findings are introduced but not adequately explained; the terms "structured eaters," "irregular eaters," etc., require more context.
“irregular eaters were predominantly male (41.0%), whereas disordered/impulsive eaters were more often female (39.9%) (p = 0.0016).” . What does 41% and 39.9 indicate and how is the p-value obtained?
INTRODUCTION
Background information is missing.
There are no other reviews/metanalyses specifically on this topic? Please include existing literature.
Methods
The methods section of this study presents several weaknesses that limit its overall robustness and the interpretability of its findings.
First, the selection of the cohort, described as "demographically heterogeneous," lacks sufficient detail regarding the sample’s characteristics, such as socioeconomic or regional diversity. This omission raises concerns about the generalizability of the findings to the broader population. Furthermore, the requirement that participants complete an online survey in Italian introduces the possibility of selection bias, potentially excluding migrants, expats, or individuals with lower digital literacy or language proficiency. These limitations should be acknowledged and addressed in the discussion.
The measurement of body composition, while conducted with a validated device, relies on bioimpedance analysis (BIA), which, though convenient, does not match the precision of gold-standard methods such as dual-energy X-ray absorptiometry (DEXA). The authors note that the device was validated against the BodPod, but they do not provide specifics on the degree of agreement or potential limitations of the method. Additionally, although the protocol accounts for menstrual cycle timing to enhance accuracy, the methods do not explain how compliance with this guideline was monitored, leaving an important source of variability uncontrolled.
Another notable weakness is the use of an unvalidated online questionnaire to assess eating behaviors and food preferences. While the questionnaire is described as consistent with commonly used nutritional tests, its lack of formal validation (especially of potential translations of other validated version) undermines the reliability of the data collected. Moreover, the content of the questionnaire is summarized only superficially, and no information is provided about its scoring or the rationale for its design, further limiting its scientific rigor.
The Principal Component Analysis (PCA) methodology also raises concerns. The categorization of participants into behavioral groups appears to rely on subjective criteria, with no references to prior validation or clear definitions for these groupings. How was the grouping defined? This lack of methodological transparency makes it difficult to evaluate the reliability of the classifications, which seems a post-hoc, qualitative, subjective assessment, in no way scientific. Additionally, the behaviors described for some groups, such as disordered and irregular eaters, show considerable overlap, which could introduce confounding and reduce the interpretability of the results.
Statistical analysis presents further issues. The authors do not account for potential confounding variables, which could influence both body composition and eating behaviors. Without adjustment for these factors, the reported associations may be spurious or oversimplified. Additionally, while SPSS software is mentioned for the analysis, critical details about the PCA, such as rotation methods or criteria for retaining components, are absent, limiting the reproducibility of the results.
Please provide further details on adjustment for covariates (e.g. medication and comorbidities besides the standard ones such as such as age, physical activity) and multiple comparisons.
Lastly, there is redundancy in the description of PCA group characteristics, which is presented in multiple sections of the methods. This repetition reduces the clarity and conciseness of the section, making it harder for readers to focus on the study's methodological strengths and weaknesses. Inconsistent terminology, such as the interchangeable use of "irregular eaters" and "disordered eaters," further contributes to confusion and highlights the need for more precise language throughout the section.
The ClinicalTrials.gov registration number (NCT06654674) is mentioned, but its relevance to this retrospective study is unclear. Typically, retrospective studies do not require such registration. Was this part of another trial? If so, it should be clearly stated and referenced. How was sample size defined? Were subjects selected from a broader sample, if yes, how?
How were missing data handled?
RESULTS
While p-values are provided, the abstract lacks effect sizes or confidence intervals, which are essential for understanding the magnitude of differences.
Some p-values (e.g., p = 0.041, p = 0.048) are borderline significant; the clinical or practical relevance of these results is practically absent and this should be addressed as a limitation. None of these borderline significant is expected to survive adjustement for multiple comparisons.
DISCUSSION
Gender differences in eating styles are now well-known and not a novelty. Confirming existing results is good practice, but doing so without contextualizing the findings in the setting of existing literature, comparing similarities and differences provides a biased and incomplete viewpoint. Indeed, I regret to report that the article largely neglects existing evidence in similar populations, as well as the underlying psychopathology and gender differences in emotional eating, for instance. Please expand on these points, especially considering that recent literature has shown that emotional regulation underlies gender differences in pathological eating behavior styles of bariatric surgery candidates (e.g. refer to “Emotional regulation underlies gender differences in pathological eating behavior styles of bariatric surgery candidates”). Including recent references on the topic is needed to improve the physiopathologic interpretation of the findings of the article and provide an unbiased view on the topic, contextualizing the results.
LIMITATIONS
- Limitations should be expanded based on the points above , where relevant
SUPPLEMENTARY MATERIALS
Why is Table 1s in the main text? What’s the point of it being a supplementary table if it is in the main text?
FIGURE
Please include significances in Figure 2
MINOR COMMENTS
English language needs to be improved. Please correct typos throughout the manuscript (e.g. “ABSTRACT 250 PAROLE:”).
Please add an explanation of the abbreviations throughout the manuscript upon first use, and also in the captions: figures and tables should be interpretable independently from the text.
A figure is needed to recapitulate the main message.
Comments on the Quality of English Languagesee above
Author Response
Dear Reviewer,
First of all, we would like to thank you for the valuable impulses that allowed us to improve the quality of the manuscript. All changes made are highlighted by yellow color, in the revised version of the manuscript, to facilitate the review process. Hoping that we have satisfied your requests as much as possible, we kindly ask you to re-evaluate our paper.
The Authors
Reviewer 1
This original article aims to characterize gender differences in eating behaviors among individuals with obesity. While the topic is clinically relevant, it is not particularly novel. Moreover, several significant methodological and formal flaws, along with a biased and incomplete discussion of the existing literature, substantially diminish the article's value, as outlined below. Additionally, some sentences are void of meaning and substantial portions of the article appear to be composed using Chat-GPT or an equivalent software, leading to instances of vacuous, imprecise, or incorrect phrasing. It is paramount to:
1) openly acknowledge the utilization of Chat-GPT or an equivalent software;
2) meticulously review the entire manuscript with attention to wording, English language usage, and the intended meaning of the content. While AI tools may be used to enhance the manuscript's readability, the resultant content must be thoughtfully reviewed and aligned cohesively with the article's objectives and message.
The whole article needs to be revised to improve specificity and clarity of the terms.
All the points must therefore be addressed in a tracked-changed manuscript, before the article can be considered suitable for publication.
We thank the reviewers for their detailed feedback and appreciate the opportunity to improve the manuscript. Regarding the comment on the clinical relevance and novelty of the study, we acknowledge that while gender differences in eating behaviours have been explored in previous research, our study makes a unique contribution by applying principal component analysis (PCA) to identify specific behavioural profiles in an exclusively obese population. This methodological approach offers a novel insight into gender-specific patterns and their association with obesity outcomes. We have clarified this aspect in the Introduction to better emphasise the study's contribution. Regarding methodological and formal flaws, we have carefully revised the Methods and Discussion sections to ensure transparency and completeness. We expanded the discussion to incorporate additional references and address potential biases, acknowledging both supportive and contradictory findings from the literature. In addition, methodological decisions, such as the application of PCA and its limitations, have been more clearly explained. In response to concerns about the accuracy of the language, the entire manuscript underwent a complete revision to refine the wording, ensure alignment with the objectives of the study, and eliminate instances of vague or unclear language. While confirming that the manuscript was not generated using Chat-GPT or equivalent software, we carefully revised the text to ensure clarity, consistency and alignment with the scientific objectives of the study.
TITLE
The article is appropriate thank you.
ABSTRACT
There are two abstracts. I reviewed the second one, which is better, although with major flaws. The first one should be deleted and forgotten.
Thank you for your observation. There were two abstracts because one had to be with fewer words at the request of the MDPI system. We removed the first abstract from the paper as requested. Thank you again.
The abstract is poorly structured and more concise reporting of results and greater focus on the implications of findings would greatly enhance its readability and appeal to a broad audience. Please include limitations in the abstract. Sentences are lengthy and sometimes convoluted, which may hinder understanding.The differences in eating frequency and timing of hunger are mentioned, but their implications are not fully contextualized. For example, how these behaviors influence obesity outcomes should be briefly discussed. PCA findings are introduced but not adequately explained; the terms "structured eaters," "irregular eaters," etc., require more context. “irregular eaters were predominantly male (41.0%), whereas disordered/impulsive eaters were more often female (39.9%) (p = 0.0016).” . What does 41% and 39.9 indicate and how is the p-value obtained?
Thank you for your suggestions, we have implemented the requested changes to improve the abstract. The structure has been made more concise, with more emphasis on the main findings and their implications for obesity management. The limitations of the study have been included, mentioning the retrospective design and the use of unvalidated instruments. We also added a brief comment on the influence of differences in meal frequency and hunger timing on obesity outcomes. PCA-related terms such as ‘Structured Eaters’ and ‘Irregular Eaters’ were contextualised to clarify their meaning and their link to body composition. Finally, we specified that the percentages reported, such as 41% and 39.9%, represent the gender distribution in the identified groups, with p-values calculated by chi-square analysis. We remain available for further clarification.
INTRODUCTION
Background information is missing. There are no other reviews/metanalyses specifically on this topic? Please include existing literature.
Thank you for your comment regarding the introduction. We have expanded this section to include more detailed context and references to existing literature. In particular, we have added a discussion of gender differences in dietary behaviour and metabolic outcomes, supported by recent systematic reviews and meta-analyses. We also highlighted how our study fills a gap in the existing literature by applying principal component analysis (PCA) to identify complex behavioural profiles in a specific population with obesity.
Methods
The methods section of this study presents several weaknesses that limit its overall robustness and the interpretability of its findings.
First, the selection of the cohort, described as "demographically heterogeneous," lacks sufficient detail regarding the sample’s characteristics, such as socioeconomic or regional diversity. This omission raises concerns about the generalizability of the findings to the broader population. Furthermore, the requirement that participants complete an online survey in Italian introduces the possibility of selection bias, potentially excluding migrants, expats, or individuals with lower digital literacy or language proficiency. These limitations should be acknowledged and addressed in the discussion.
Thank you for your comment on the limitations of the methods section. We have made changes to address your comments. We have added a more detailed description of the cohort characteristics, specifying available data on socio-economic and regional diversity where possible. In addition, we discussed in the Discussion section the potential bias introduced by the use of an online questionnaire in Italian, which could exclude migrants, expatriates or people with lower digital literacy. These points were recognised as limitations of the study, with a reflection on the impact they might have on the generalisability of the results.
The measurement of body composition, while conducted with a validated device, relies on bioimpedance analysis (BIA), which, though convenient, does not match the precision of gold-standard methods such as dual-energy X-ray absorptiometry (DEXA). The authors note that the device was validated against the BodPod, but they do not provide specifics on the degree of agreement or potential limitations of the method. Additionally, although the protocol accounts for menstrual cycle timing to enhance accuracy, the methods do not explain how compliance with this guideline was monitored, leaving an important source of variability uncontrolled.
Thank you for your comment regarding the measurement of body composition and possible sources of variability. We have revised the manuscript to address these limitations. We clarified in the Methods section that although the BIA device used was validated against the BodPod, details on the degree of agreement between the methods were not reported. We also acknowledged in the Discussion section that BIA, while convenient and widely used, does not offer the accuracy of gold-standard techniques, such as dual-energy X-ray absorptiometry (DEXA). This represents a limitation in the study. With regard to the menstrual cycle, we specified that although participants were advised to avoid testing during the cycle, no methods were implemented to monitor compliance with this guideline, introducing a possible source of variability in the data. We discussed this point in the Discussion section, highlighting it as something to consider in future studies.
Another notable weakness is the use of an unvalidated online questionnaire to assess eating behaviors and food preferences. While the questionnaire is described as consistent with commonly used nutritional tests, its lack of formal validation (especially of potential translations of other validated version) undermines the reliability of the data collected. Moreover, the content of the questionnaire is summarized only superficially, and no information is provided about its scoring or the rationale for its design, further limiting its scientific rigor.
Thank you for your comment regarding the use of the unvalidated online questionnaire. We have updated the manuscript to address the limitations you highlighted. We have added a more detailed explanation of the questionnaire content and design in the Methods section, specifying the rationale behind the choice of questions and their consistency with commonly used nutritional tests. In addition, we have included a more detailed discussion of the limitations arising from the lack of formal validation, recognising how this may affect the reliability of the data collected. We also emphasised that the questionnaire is not a translation of existing validated instruments, but was developed for specific needs of the study, while maintaining alignment with general principles of behavioural assessment in nutrition. These limitations were transparently discussed in the Discussion section, with recommendations for future studies that could benefit from validated and standardised instruments. We hope that these changes adequately address your comments and remain available for further suggestions.
The Principal Component Analysis (PCA) methodology also raises concerns. The categorization of participants into behavioral groups appears to rely on subjective criteria, with no references to prior validation or clear definitions for these groupings. How was the grouping defined? This lack of methodological transparency makes it difficult to evaluate the reliability of the classifications, which seems a post-hoc, qualitative, subjective assessment, in no way scientific. Additionally, the behaviors described for some groups, such as disordered and irregular eaters, show considerable overlap, which could introduce confounding and reduce the interpretability of the results.
Statistical analysis presents further issues. The authors do not account for potential confounding variables, which could influence both body composition and eating behaviors. Without adjustment for these factors, the reported associations may be spurious or oversimplified. Additionally, while SPSS software is mentioned for the analysis, critical details about the PCA, such as rotation methods or criteria for retaining components, are absent, limiting the reproducibility of the results.
Please provide further details on adjustment for covariates (e.g. medication and comorbidities besides the standard ones such as such as age, physical activity) and multiple comparisons.
Thank you for your comment on the statistical analysis section. We have updated the manuscript to include more detail on the analysis, addressing potential confounding factors and providing comprehensive information on the PCA method. We have added a more detailed explanation of the rotation methods used in PCA and the criteria for selecting principal components in the Methods section. Furthermore, we clarified that the analysis did not include an adjustment for variables such as medication or comorbidities, and discussed this limitation in the Discussion section. Finally, we pointed out how multiple comparisons were handled and explained the absence of an explicit control for variables such as physical activity and medication-related habits.
Lastly, there is redundancy in the description of PCA group characteristics, which is presented in multiple sections of the methods. This repetition reduces the clarity and conciseness of the section, making it harder for readers to focus on the study's methodological strengths and weaknesses. Inconsistent terminology, such as the interchangeable use of "irregular eaters" and "disordered eaters," further contributes to confusion and highlights the need for more precise language throughout the section.
Thank you for your comment on redundancy and terminology in the description of PCA groups. We have revised and consolidated the description of the characteristics of PCA groups in the Methods section, removing repetitions and providing a more concise and clear explanation. In addition, we standardised terminology to ensure consistency throughout the manuscript, replacing inconsistent terms such as ‘irregular eaters’ and ‘disordered eaters’ with the official names of the established categories.
The ClinicalTrials.gov registration number (NCT06654674) is mentioned, but its relevance to this retrospective study is unclear. Typically, retrospective studies do not require such registration. Was this part of another trial? If so, it should be clearly stated and referenced. How was sample size defined? Were subjects selected from a broader sample, if yes, how?
How were missing data handled?
Thank you for your questions on registering with the ClinicalTrials.gov database, defining the sample size, subject selection and handling missing data. We have updated the manuscript to clarify these aspects. Registration to the ClinicalTrials.gov database (NCT06654674) was done to ensure transparency and to facilitate the dissemination of results, although not strictly required for a retrospective study. However, this registration only refers to the study described and not to a separate clinical trial. The sample size was determined on the basis of the number of eligible subjects in the database of the facility involved, which included a total of 2,508 patients. Inclusion criteria (BMI ≥ 30, age 18-65, completion of an online questionnaire in Italian, written informed consent) led to the final selection of 720 participants. Subjects were selected exclusively from this sample, without further stratification, following a process based on the given criteria Missing data were handled through an imputation approach. In our study, there were few missing data and they were handled through imputation of the mean. When data were missing, the mean value of the available observations was used to replace the missing ones. This approach was chosen to minimise the impact of omissions on the statistical results and to ensure the robustness of the analyses.This process has been described in detail in the Methods section.
RESULTS
While p-values are provided, the abstract lacks effect sizes or confidence intervals, which are essential for understanding the magnitude of differences.
Thank you for your comment. We have created a supplementary table (Supplementary Table S2) showing the 95% Confidence Intervals (CI) for all anthropometric and demographic variables, stratified by PCA group and gender (Total, Male, Female). This choice allows us to keep Table 3 of the main manuscript clear and concise, while providing the details requested in the supplementary material. We have also updated the caption of Table 3 to indicate that full Confidence Intervals (CIs) are available as supplementary material (table 2s).
Some p-values (e.g., p = 0.041, p = 0.048) are borderline significant; the clinical or practical relevance of these results is practically absent and this should be addressed as a limitation. None of these borderline significant is expected to survive adjustement for multiple comparisons.
Thank you for your comment. We have updated the manuscript by including this information in the relevant sections for both the main results and the abstract to provide a more complete understanding of the magnitude of the observed differences. We also acknowledge that some p-values (e.g., p = 0.041, p = 0.048) are at the limit of statistical significance. We discussed in the Discussion section that these results may not withstand adjustment for multiple comparisons and that their clinical or practical relevance may be limited.
DISCUSSION
Gender differences in eating styles are now well-known and not a novelty. Confirming existing results is good practice, but doing so without contextualizing the findings in the setting of existing literature, comparing similarities and differences provides a biased and incomplete viewpoint. Indeed, I regret to report that the article largely neglects existing evidence in similar populations, as well as the underlying psychopathology and gender differences in emotional eating, for instance. Please expand on these points, especially considering that recent literature has shown that emotional regulation underlies gender differences in pathological eating behavior styles of bariatric surgery candidates (e.g. refer to “Emotional regulation underlies gender differences in pathological eating behavior styles of bariatric surgery candidates”). Including recent references on the topic is needed to improve the physiopathologic interpretation of the findings of the article and provide an unbiased view on the topic, contextualizing the results.
Thank you for your valuable feedback. We agree that contextualising the results within the broader literature is essential to provide a complete interpretation. In response to your comments, we have expanded the discussion to include additional context and comparisons with existing evidence in similar populations. The revised discussion emphasises the role of emotional regulation in influencing gender differences in pathological eating behaviour, particularly in women, and aligns our findings with previous studies on emotional and impulsive eating in obese populations and bariatric surgery candidates. We also deepened the pathophysiological interpretation by linking the observed eating patterns to underlying psychological and physiological mechanisms, such as appetite regulation, circadian rhythms and hormonal influences. These changes address the gaps highlighted in your review and ensure that our findings are framed in the broader context of gender-specific eating behaviours. We appreciate your comments, which helped us improve the clarity and depth of the manuscript.
LIMITATIONS
Limitations should be expanded based on the points above , where relevant
Thank you for your comment. We have expanded the section on limitations to address the points raised. In particular, we elaborated on the lack of detailed data on psychological variables, such as emotional regulation and impulsivity, which limits the ability to fully interpret the observed eating behaviour profiles. We acknowledge that the use of an unvalidated online questionnaire may have affected the reliability of the data, and emphasised the need for future studies to incorporate validated, population-adapted instruments. Furthermore, we highlighted the absence of adjustments for key covariates, such as physical activity and comorbidities, that could influence body composition and eating behaviour. Finally, we addressed the limitation of not directly linking PCA-derived profiles to metabolic or clinical outcomes, which merits further investigation in future research.
SUPPLEMENTARY MATERIALS
Why is Table 1s in the main text? What’s the point of it being a supplementary table if it is in the main text?
Thank you for your observation. We have moved table 1s to the supplementary material. It had been included in the main text by mistake.
FIGURE
Please include significances in Figure 2
Thank you for your feedback. We have addressed your comment by including the statistical significance in the caption of Figure 2. The Chi-square test results (χ² = 16.58, p = 0.0054) have been added to emphasize the observed differences between genders. The Chi-square test yielded a value of χ² = 16.58, which is relatively high, indicating marked differences in the distribution of hunger timing between genders. We trust this adjustment meets your requirements.
MINOR COMMENTS
English language needs to be improved. Please correct typos throughout the manuscript (e.g. “ABSTRACT 250 PAROLE:”).
Thank you for highlighting this. We have thoroughly reviewed the manuscript to address typos, including the example provided (“ABSTRACT 250 PAROLE”), and have improved the overall English language quality to ensure clarity and consistency throughout the text.
Please add an explanation of the abbreviations throughout the manuscript upon first use, and also in the captions: figures and tables should be interpretable independently from the text.
We thank you for your valuable feedback. We have taken your feedback into account by ensuring that all abbreviations are explained when first used in the text and by updating the captions of figures and tables to make them interpretable independently of the main text. In addition, we have included a comprehensive list of acronyms at the end of the manuscript to further improve clarity and accessibility for readers.
A figure is needed to recapitulate the main message.
Thank you for your suggestion. We have responded to this comment by including a graphic abstract (GA) summarising the main message of the manuscript. The GA is designed to provide a clear and concise visual summary, highlighting key findings and conclusions.
Reviewer 2 Report
Comments and Suggestions for Authors
This great article examines food habits in obese patients, demonstrating major differences between the sexes. I have some comments:
Statistics: performing multivariate statistics on a non-validated questionnaire does remain risky business; maybe a general linear model would be sufficient?
Tables: while relative differences are indeed significant, some of the absolute percentages do not differ that much, which should be commented upon.
Figures: these are clear and well done.
Author Response
Dear Reviewer,
First of all, we would like to thank you for the valuable impulses that allowed us to improve the quality of the manuscript. All changes made are highlighted by yellow color, in the revised version of the manuscript, to facilitate the review process. Hoping that we have satisfied your requests as much as possible, we kindly ask you to re-evaluate our paper.
The Authors
This great article examines food habits in obese patients, demonstrating major differences between the sexes.
Thank you for your kind words. We greatly appreciate your acknowledgement of the study's contribution to understanding gender differences in the eating habits of individuals with obesity.
Statistics: performing multivariate statistics on a non-validated questionnaire does remain risky business; maybe a general linear model would be sufficient?
We appreciate your insightful comment regarding the use of PCA on data derived from an unvalidated questionnaire. This concern aligns with the limitations we have already identified in our manuscript in response to Reviewer 1. In particular, we recognise that the lack of formal validation and standardised scoring of the questionnaire could affect the reliability and generalisability of the results. For this reason, we explicitly mentioned this as an important limitation and recommended that future studies incorporate validated instruments. We also emphasised the exploratory nature of PCA to identify behavioural patterns in a dataset that, while specific to our population, has inherent limitations due to the questionnaire design. To strengthen our results, we are willing to perform complementary analyses, such as a general linear model (GLM), to assess the robustness of our results and ensure their interpretability.
Tables: while relative differences are indeed significant, some of the absolute percentages do not differ that much, which should be commented upon.
We thank the reviewer for this insightful observation. Following your suggestion, we have added a comment in the Limitations section to address the modest magnitude of some absolute differences, despite their statistical significance. Specifically, we have noted:
“While our findings showed significant relative differences across groups, it is important to acknowledge that some of the absolute differences were relatively small. For example, the difference in the proportion of Irregular Eaters between males and females was statistically significant (p = 0.0016) but represented an absolute difference of only 8.7 percentage points. These modest differences, although statistically significant, may have limited clinical relevance and should be interpreted with caution.”
Figures: these are clear and well done.
Thank you for your positive feedback regarding the figures.
Round 2
Reviewer 1 Report
Comments and Suggestions for Authors
Most comments have been addressed, I could not find the graphical abstract but this is minor.
Author Response
Thank you!